# On the fractal patterns of language structures

**Leonardo Costa Ribeiro**[1]*, **Américo Tristão Bernardes**[2], **Heliana Mello**[3]

**1** Departamento de Ciências Econômicas, Faculdade de Ciências Econômicas, Universidade Federal de Minas Gerais, Belo Horizonte, Minas Gerais, Brasil, **2** Departamento de Física, Instituto de Ciências Exatas e Biológicas, Universidade Federal de Ouro Preto, Ouro Preto, Minas Gerais, Brasil, **3** Faculdade de Letras, Universidade Federal de Minas Gerais, Belo Horizonte, Minas Gerais, Brasil

☯ These authors contributed equally to this work.
* lcr@cedeplar.ufmg.br

**Data Availability Statement:** All relevant data are within the manuscript and its Supporting information files.

**Funding:** This work was partly supported by the Brazilian agencies CNPq (307633/2019-5 and 312020/2021-0) and PRPq-UFMG. There was no

## Abstract

Natural Language Processing (NLP) makes use of Artificial Intelligence algorithms to extract meaningful information from unstructured texts, i.e., content that lacks metadata and cannot easily be indexed or mapped onto standard database fields. It has several applications, from sentiment analysis and text summary to automatic language translation. In this work, we use NLP to figure out similar structural linguistic patterns among several different languages. We apply the word2vec algorithm that creates a vector representation for the words in a multidimensional space that maintains the meaning relationship between the words. From a large corpus we built this vectorial representation in a 100-dimensional space for English, Portuguese, German, Spanish, Russian, French, Chinese, Japanese, Korean, Italian, Arabic, Hebrew, Basque, Dutch, Swedish, Finnish, and Estonian. Then, we calculated the fractal dimensions of the structure that represents each language. The structures are multi-fractals with two different dimensions that we use, in addition to the token-dictionary size rate of the languages, to represent the languages in a three-dimensional space. Finally, analyzing the distance among languages in this space, we conclude that the closeness there is tendentially related to the distance in the Phylogenetic tree that depicts the lines of evolutionary descent of the languages from a common ancestor.

## Introduction

When we discuss the emergence of language in the human species, different theories come to the fore. It is not possible to discuss language evolution outside the evolutionary history of humankind. Studies that focus on language evolution take into account different types of evidence from archeology, genetics, anthropology, language acquisition, among other fields. There is no simple, much less easy, answer to the issue of language evolution.

As of recently, the gestural hypothesis [1] started gaining momentum over the well-known genetic mutation hypothesis [2]. The former predicts a gradual development of gestural communication, which precedes Homo sapiens. The latter, on the other hand, proposes a single recent genetic mutation in sapiens as the beginning of language for humanity.

additional external funding received for this study. The funders had no role in study design, data collection and analysis, decision to publish, or preparation of the manuscript.

**Competing interests:** The authors have declared that no competing interests exist.

Whatever the answer to the fascinating question of how language emerged, the fact is that today there are more than 7,000 languages spoken or signed in the world, and every human society known has language and every human child, except very few out of normality cases, will develop language at a tender age. However, what concerns us in this paper is not the issue of language evolution as a species trait, but rather that of similarity among languages, approached from a computational perspective.

Ever since comparative linguistics emerged as a discipline (initially as comparative philology), scholars have been trying out methodologies that support similarity comparisons among languages. There are different processes through which languages may resemble each other, notably genetic relationships, i.e., a common ancestral language, which is the major focus of comparative linguistics; language contact through borrowing and shift; the spread of areal features; universal tendencies, such as pointed out in typological studies, or even sheer coincidence in isolated word forms. Whatever the reason for similarities among languages, the conduit for them is human communication and the transmission of language from generation to generation, as well as through some form of population contact leading to language contact.

Computational methods for the study of language similarities have been gaining ground in the past twenty years due both to the buildup of large data sets and the development of computational methods and tools. The reliability of automated methods for the analysis of historical linguistic data is still disputed and solutions such as computer-assisted language comparisons have been proposed [3]. Computational methodology for language similarities has been explored mostly within the contrastive historical and the typological fronts. While the former has a focus on cognate forms as a basic tenet [4], the latter explores language structure in its broadest sense, involving grammatical aspects ranging from syntax to phonology [5].

Among the many methods presently in use, phylogenetic methods have gained a boost due to the transfer of ideas and software from computational biology and to the release of several large electronic data resources suitable for systematic comparative work. These methods typically involve the construction of language family trees, which illustrate the historical relationships between languages and the order in which they diverged from a common ancestral language [6]. These trees are often constructed using statistical models and algorithms that compare linguistic features across different languages, such as sound changes, word order, and vocabulary. Some common methods used in phylogenetic linguistics include: maximum likelihood estimation, Bayesian inference, neighbour-joining. The application of computational phylogenetic methods to language change in recent times makes it possible to analyze large quantities of data employing techniques akin to a large-scale quantitative rendition of small-scale comparative studies of early historical linguistics and beyond. Among the results achieved through such methods are automatic assessment of genetic relatedness, automatic cognate detection, phylogenetic inference and ancestral state reconstruction.

McMahon & McMahon [7] is a good example of a book that introduces some of the most important areas of quantitative research in historical linguistics. The authors explore several methods available and also indicate their usefulness. As investigated by [8], the potential of computational methods for studies related to language evolution and change are significant, as they deal with different issues that have been at the core of the subject for decades. Among them, they are able to infer ancestral genealogy of a group of languages, point to answers related to the classification and expansion of language families, and investigate patterns of lexical borrowing, besides factors affecting rates of language evolution.

Since the pioneering work of Zipf [9, 10] about 90 years ago, a good deal of effort has been applied to attaining a better understanding of the basic mathematical structure of languages. Much of the research in this field studied the structure of languages according to the sequence

of words. As we shall discuss below, sequences of symbols can be interpreted as time series and many different methods derived from physics, mathematics and computation can be applied.

Languages are not merely a collection of words, but networks which can be displayed in n-dimensional spaces. Each word has a position vector in this space, and the closeness of those points is given by the context in which the words appear in the language. Depending on the distance from which we view this space, we can observe sequence of points aligned as thin thread, clusters or blurs. If we come very close, we will see points. From afar, a blur. Those structures can be studied through tools of fractal geometry. It is important to mention that, since the work of Mandelbrot [11, 12] a lot of attention has been paid to understanding the fractal structure of languages.

In the present paper, different from traditional studies, we seek this fractal structure in an n-dimensional space. Starting from the Corpora of many languages, we applied an algorithm developed in the last decade to create this space. We used the word2vec because the vector representation it creates for the words brings their meaning. So, a general analysis for this space will consider a broader linguistic aspect of the language and not only how the words are formed as a sequence of letters as in previous works in the literature. After this, we applied the tools of fractal geometry to observe similarities and differences among the many languages. It is done by identifying the languages that appear in the same cluster (similar languages) or not (not so similar languages) in the space we create from the fractal dimension of the languages and their type/token ratio. This paper is organized as follows. In the next section, we discuss the trajectory of the mathematical treatment of language, bringing up certain aspects and discussing some unresolved questions. After this, we introduce aspects of fractal geometry, with clarification for those who are not familiar with the field. Subsequently, the basic method of creating the n-dimensional space is discussed and the data base is described. Applying the methodologies we have presented, our results are shown and discussed. And at the end, we present our conclusions.

## Fractals

### Investigation of languages through fractal structure

Complex systems have received attention since the second half of the 20th century. Some characteristics like self-similarity, the emergence of patterns, and dynamical behaviours which can not be defined from the individual behaviour of its components are the signatures of these systems. The phenomena described by complex systems are observed in many fields of human knowledge, not only in natural sciences but also in social sciences, Arts and Literature. With the advent of computers, some of these properties could be simulated in simple models, like the Game of Life [13].

Human Languages, as discussed previously, show many different characteristics among them, but the study of properties which are today designed for complex systems started before the term complex system was coined.

One of the first attempts to understand the structure of a language was the mapping of word frequency [9, 10]. It observed a scale-free behaviour, which came to be known as Zipf's Law: some few words appear with a higher frequency, many others with lower. The mathematical function which describes this feature is a power-law function. A fascinating aspect is that this kind of behaviour is observed in many other systems, either natural or even artificial, like the internet [14], or earthquakes or city sizes distribution; from stock markets to scientific citations. For a review of power laws in natural phenomena, see [15].

The importance of this finding has been questioned since a "language" created at random also showed a power-law behaviour [11, 16]. However, later studies showed that human

languages have long-range correlations, i.e., they show a correlation among words or structures that are far from each other in the sentences and can not be represented by a random walk [17, 18], i.e. a path created by a succession of random steps that do not consider the information of the previous step to select the following step. For a more comprehensive discussion about this topic, see [19]. It is worth of mention that Emile Borel [20], in 1913 originally put forth the statistical problem of generating an understandable language from a random choice of letters which will form words. This was later addressed by Arthur Eddington [21], and is a classical problem in Statistical Physics.

A language is not a mere collection of words. It is defined by multiple relations among words, i.e. defined by its grammar. The grammatical rules create networks among words. One would believe that these networks may be scale-free and their parameters could reveal universalities in every human language [22–24]. The language diversity in relation to their coverage area and population also present power-law behaviour [25–28].

The study of languages from a fractal approach started in the '70, soon after Mandelbrot published his book [12]. He affirmed that, with other natural processes, languages should present a fractal structure. Shannon [29] put forward that languages should be scale-invariant, another fractal feature. Some studies seeking to define a more formal vision of the fractal structure of languages have been developed by many authors [30, 31].

In most of the studies carried out in the last decades, sentences, as time series, allow the calculation of the Hurst [32] exponent. The Hurst exponent is a mathematical measure that characterizes the long-term correlation or persistence of a time series or signal and it is related to their fractal dimension. The Hurst exponent is a quantity that ranges between 0 and 1, with values closer to 0 indicating anti-persistent behaviour (i.e., a tendency for a time series to reverse its direction frequently) and values closer to 1 indicating persistent behaviour (i.e., a tendency for a time series to continue in the same direction). A Hurst exponent of 0.5 indicates random behaviour, with no long-term memory or persistence. Montemurro [17] showed that the exponent obtained by considering the sequence of words in original texts is different from that obtained in a shuffled series of words. For original texts, this exponent is different from 0.5, which denotes that there are structures with long-range correlations. Moreover, they assume that words are the fundamental "atoms" to study those structures. Analogous efforts have been made to study the different importance of words in texts [33]. Shimizu [34] has compared results obtained in English and Japanese with profiles randomly produced and those written by patients with schizophrenia. They have also observed results confirming long-range correlation. Even in this case, the series is formed by the word sizes, which avoid grammatical or syntactical effects; one observes the difference between original texts and random ones [35].

## Fractal and multi-fractal definition

To further understand the fractals, it is necessary to comprehend what the dimension of an object means. So, consider a segment line and split it in half (see the left part of Fig 1). Considering an imaginary and homogeneous mass distribution for the line, we will obtain two smaller versions of the original segment with half its length and half its mass. Therefore, the smaller version is the original segment rescaled by a factor of 1/2 that can be written as $M_{split} = \left(\frac{1}{2}\right)^1 M$ where $M_{split}$ is the mass after the division and $l$ the original mass of the segment.

Similarly, consider now a squared area and also split each of its sides in half (see the left centre of Fig 1). This time, we will obtain four smaller squares that we need to add up to recover the original mass. So, each rescaled piece has a mass of 1/4 of the original square. Then, when it is rescaled by a factor of 1/2 its mass is rescaled by a factor of 1/4, i.e., $M_{split} = \frac{1}{4} M = \left(\frac{1}{2}\right)^2 M$.

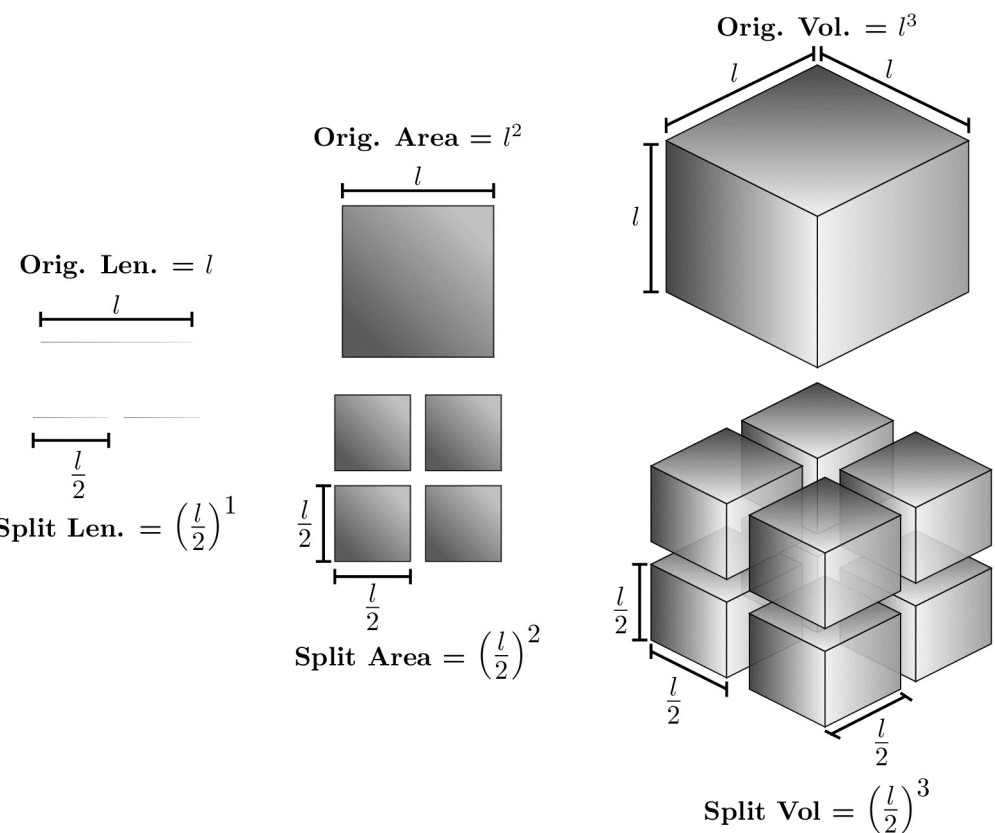

**Fig 1. Integer-dimension objects.**

Finally, let us consider a cube and split each side in half (see the right part of Fig 1). We will obtain eight smaller cubes, each with an eighth of the original mass. So, when a cube is rescaled by a factor of 1/2 its mass is rescaled by a factor of 1/8, i.e., $M_{split} = \frac{1}{8}M = \left(\frac{1}{2}\right)^8 M$

Putting all equations together to connect the rescaled mass with the rescaling factor, we obtain:

$$M_{split} = s^D M \qquad (1)$$

where $s$ is the rescaling factor, and $D$ is the object dimension. Therefore, the dimension of an object is related to how it fills space when we rescale it.

However, following this dimension definition, it is possible to construct objects with a non-integer dimension. Thereunto, start with a line segment, split it into three pieces of the same length, remove the intermediate segment, and substitute it for two segments, each showing the same length as the previous, setting up two sides of an equilateral triangle. Repeat those steps—infinite times—for each line segment of the obtained object. Fig 2 illustrate this procedure to build the Koch curve.

Due to how the Kock curve is built, if we rescaled it by a factor of 1/3 we obtain a quarter of the original curve, therefore, $M_{split} = \frac{1}{4}M = \left(\frac{1}{3}\right)^d M$. As the mass $M$ is not null, we write $3^d = 4$ or, getting the log of both sides, $d = \frac{\log(4)}{\log(3)}$. Therefore, the dimension of the Koch Curve is 1.26 that is a non-integer number.

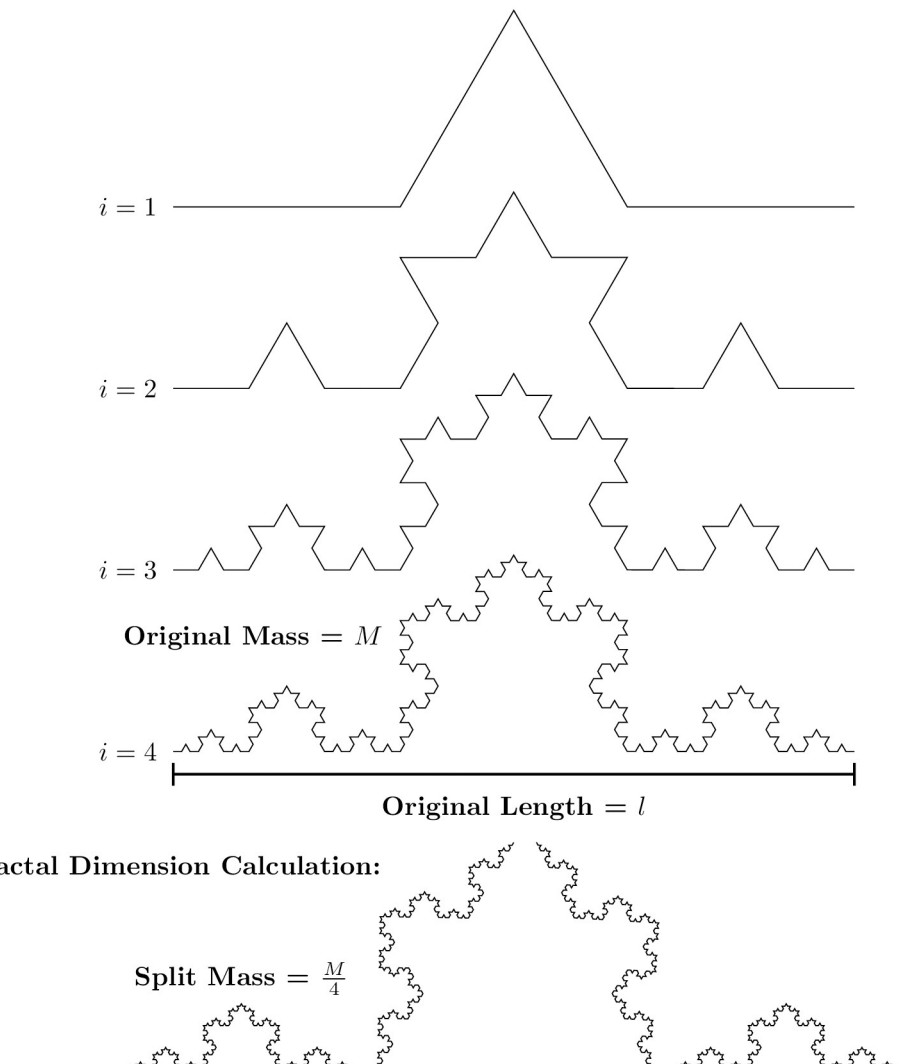

**Fig 2. Fractional-dimension object: Koch curve.**

## Interpretation of the fractal structure

So, the fractal dimension (non-integer one) is related to how the fractal curve rescales when we see it on different scales. It leads us to the self-similarity property of fractals which says that the characteristics of fractals are the same when we zoom them in or out. Thus, fractals must show a long-range spatial correlation that enables them to relate their parts with the curve as a whole. Consequently, the fractal dimension is a property of the entire curve created by the relation of the curve in different scales and not just a characteristic of a specific scale. This fact is essential for interpreting our results on the language structures, and we will return to further on.

## Creating a vector representation for words

### Where the meaning of words is

"You shall know a word by the company it keeps" [36]. This phrase by British linguist John Rupert Firth draws attention to the fact that we can get the meaning of a word by glancing at other words used in the same context in general sentences. It means that words that can be interchangeable in a sentence have similar meanings. It occurs because there is a high conditional probability of one of those words appearing in a sentence, given that the other word also appears.

### Word2vec algorithm

Considering the above, it is rather likely that the words sky and blue (or any other word related to this meaning) will appear together in a sentence. Therefore, it reflects the conditional probability of blue appearing in a sentence given that the word sky has already appeared on it (mathematically written as $P$(blue|sky)). From a vast volume of text, called corpus, it is possible to empirically calculate this conditional probability of sky and blue appearing in the same sentence. Likewise, it is also possible to calculate the conditional probability for all other pairs of words that appear in the corpus.

Then, we build the word2vec algorithm as a three-layered neural network that we train using the empirical conditional probabilities [37]. In the neural network, either the input or output layers are the one-hot representation of all words contained in the corpus. Thus, each neuron in these layers is associated with a specific word. E.g., to represent a word as input or output of the network, we use a vector filled up with zeros except in the entry equivalent to the neuron representing the referred word where we fill in with a one, see Fig 3.

Then, we train the neural network, i.e., calculate the coefficients associated with each link among neurons in order for the neural network output to reproduce, as close as possible, the conditional probability of each pair of words appearing in the same sentence. Thereby, each output neuron will be associated to $n$ coefficients—one for each intermediate neuron, as Fig 3 shows. Finally, we build an $n$-dimensional space where each dimension represents the coefficient obtained in each intermediate neuron. So, we can represent a specific word in this space as a vector whose elements are the coefficients associated with the intermediate layer.

For example, the word *sky*, illustrated in Fig 4, will become the vector [0.3, 0.2, 0.5, . . ., 0.9]. And the same can be done with all other words of the corpus.

Therefore, creating a vector representation for the words that maintain their meaning relationship is possible once we preserve the conditional probability of each pair appearing in a sentence because we use this probability to train the neural network.

The number of neurons of the intermediate layer defines the dimension of the space created to represent the words. Ideally, this dimension is much smaller than the number of neurons of the input or output layer, which is the words' one-hot representation. So, the word2vec is an embedding operation that reduces the dimension of space that we use to represent the word, commonly, from a hundred thousand (the number of different words of the corpus) to some hundreds (number of intermediate neurons). In addition, this reduction preserves the meaning relation among the words.

## Data and methodology

### CoNLL 2017 corpora

To get the corpora of several languages to carry out our analysis, we have the data provided by The Conference on Computational Natural Language Learning (CoNLL) from 2017 [38]. The

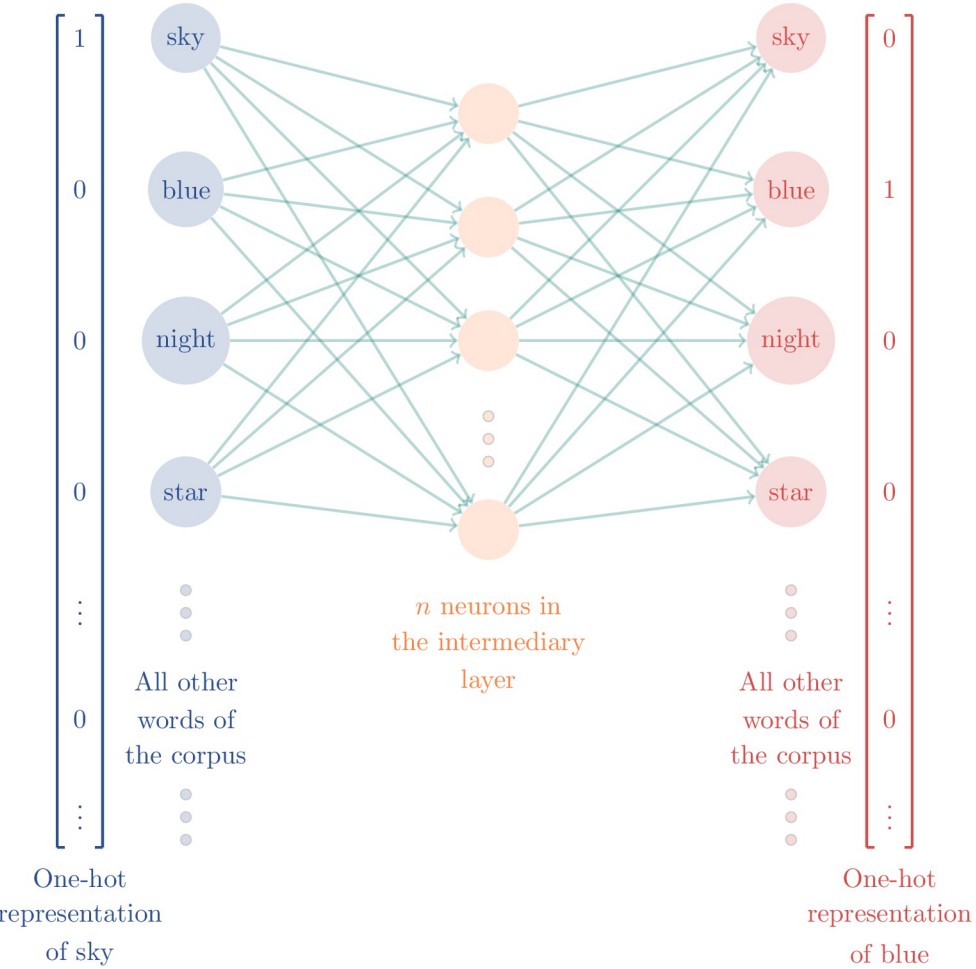

**Fig 3. Word2vec neural network.**

CoNLL 2017 featured a shared task to train and test their machine learning application on the same data sets focused on dependency parsers for several languages using real-world datasets.

As raw data from the languages, the CoNLL 2017 used the publicly available web crawl created and maintained by the non-profit CommonCrawl Foundation [39–41]. The data is publicly available in the Amazon cloud and were collected from many independent crawls from 2008 to 2017. As a reference of the magnitude of the raw data used by December 2017, the CommonCrawl provided around 2.9 billion pages totalizing around 240 TiB of information.

The participants processed those raw texts to identify the text language and generate automatic tokenization, segmentation, morphological annotations, and dependency trees. The CLD2 was used as the language detector. CLD2 is a Naïve Bayesian classifier and probabilistically detects over 80 languages in Unicode UTF-8 text [42].

Furthermore, CoNLL 2017 Corpora also provides an annotation scheme based on (universal) Stanford dependencies, Google universal part-of-speech tags, and the Interset interlingua for morphosyntactic tagsets [38].

Focusing on the part of CoNLL 2017 that is effectively used in this work, Table 1 shows the number of tokens—number of words in the corpus regardless of their repetition—and types—

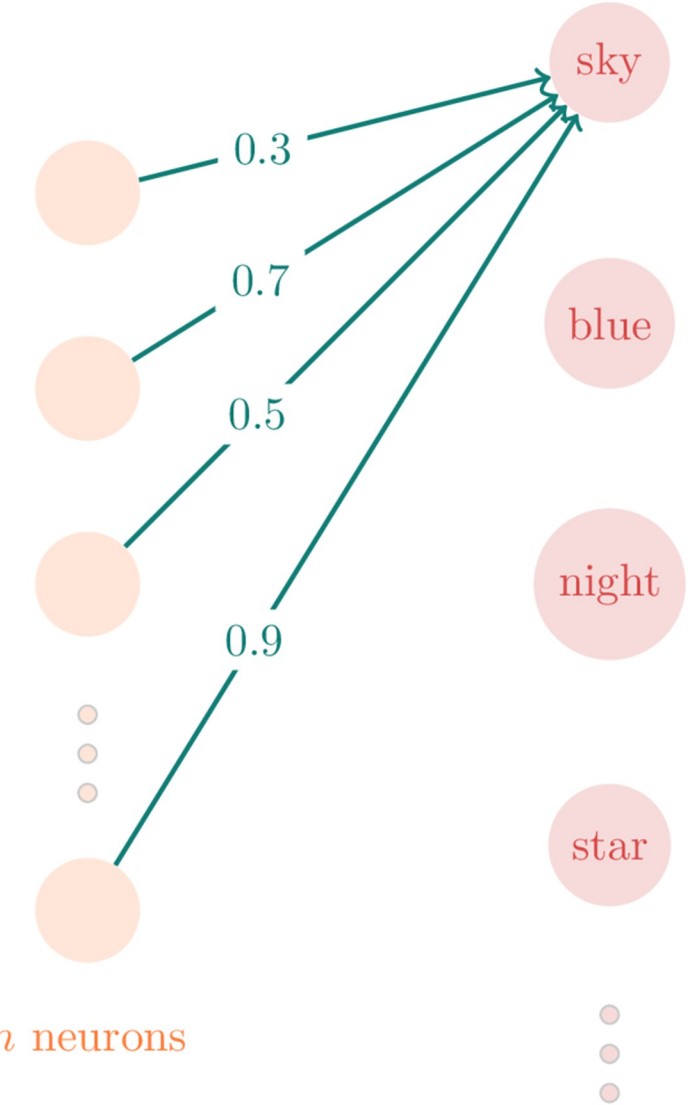

**Fig 4. Vector representation of words by word2vec.**

number of distinct words in the corpus—on CoNLL 2017 of each language analyzed in this work. The number of tokens and types is considerably high for all languages providing a statistical representativity to our analysis. Also, regarding representativity, we have chosen languages from several different branches in the Indo-European tree in addition to non-Indo-European languages, such as Estonian and Finnish (Finno-Uralic), Basque, Korean, Chinese, Japanese, Arabic and Hebrew.

The dimension of language samples taken into account in our analysis (see Table 1), in which English is the most extensive corpus and Basque is the smallest one, with all other corpora showing the scale of millions of words, thus rendering the application of the word2vec algorithm possible. So, we used 100-dimensional word2vec vector space of the selected languages available on NLPL Repository (http://vectors.nlpl.eu/reposito.ry/) [43] with algorithm "word2vec continuous skipgram. Thus, we calculated the fractal dimension of the languages' structure as follows.

**Table 1. Number of tokens and types on CoNLL 2017 of each language analyzed in this work.**

| Language | Tokens (million) | Types (million) |
|---|---|---|
| English | 9,974 | 4.03 |
| Portuguese | 6,173 | 2.54 |
| German | 6,298 | 4.95 |
| Spanish | 5,968 | 2.66 |
| Russian | 3,386 | 3.34 |
| French | 5,495 | 2.57 |
| Chinese | 1,608 | 1.94 |
| Japanese | 5,459 | 3.99 |
| Korean | 552 | 1.78 |
| Italian | 5364 | 2.47 |
| Arabic | 1,009 | 1.07 |
| Hebrew | 643 | 0.67 |
| Basque | 165 | 0.43 |
| Dutch | 3,075 | 2.61 |
| Swedish | 3,101 | 3.01 |
| Finnish | 1,053 | 2.43 |
| Estonian | 342 | 0.93 |

## Calculating the fractal dimension by box counting

Since we have now the languages represented as a point distribution on a particular space, we can analyze how those points (words) fill the space by analyzing the fractal dimension of the structure (distribution) created (see section Fractals).

To calculate the fractal dimension, we use the box-counting method that divides the space filled by the fractal in equal length segments forming, in our case, a hypercube of 100-dimension. Then, we count the number of hypercubes that the structures go through. For an illustration of a bi-dimensional case of the Koch curve, please, see Fig 5. If we associate an arbitrary and imaginary mass density to the structure, the total number of filled boxes will be approximately the structure's mass. Afterwards, we vary the side length $\delta$ of the hypercubes from an initial value $\delta_0$ until a maximum values $\delta_f$ and count the number of boxes filled in each value of $\delta$. Changing the $\delta$ is equivalent to altering the scales used to analyze the structure or, in

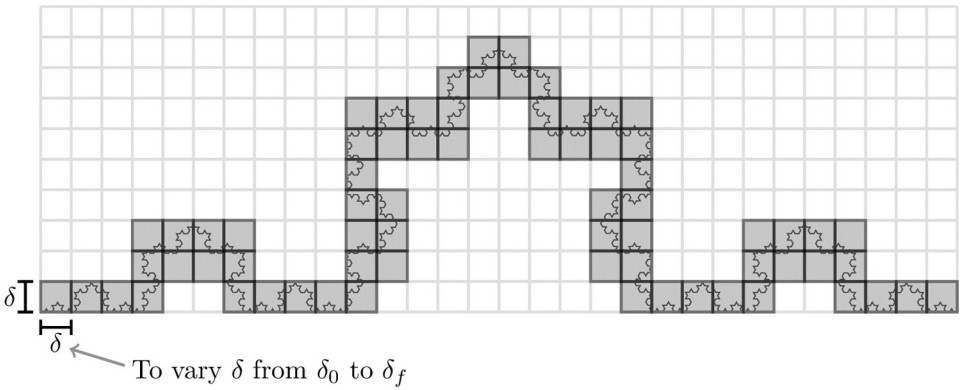

**Fig 5. Box counting.**

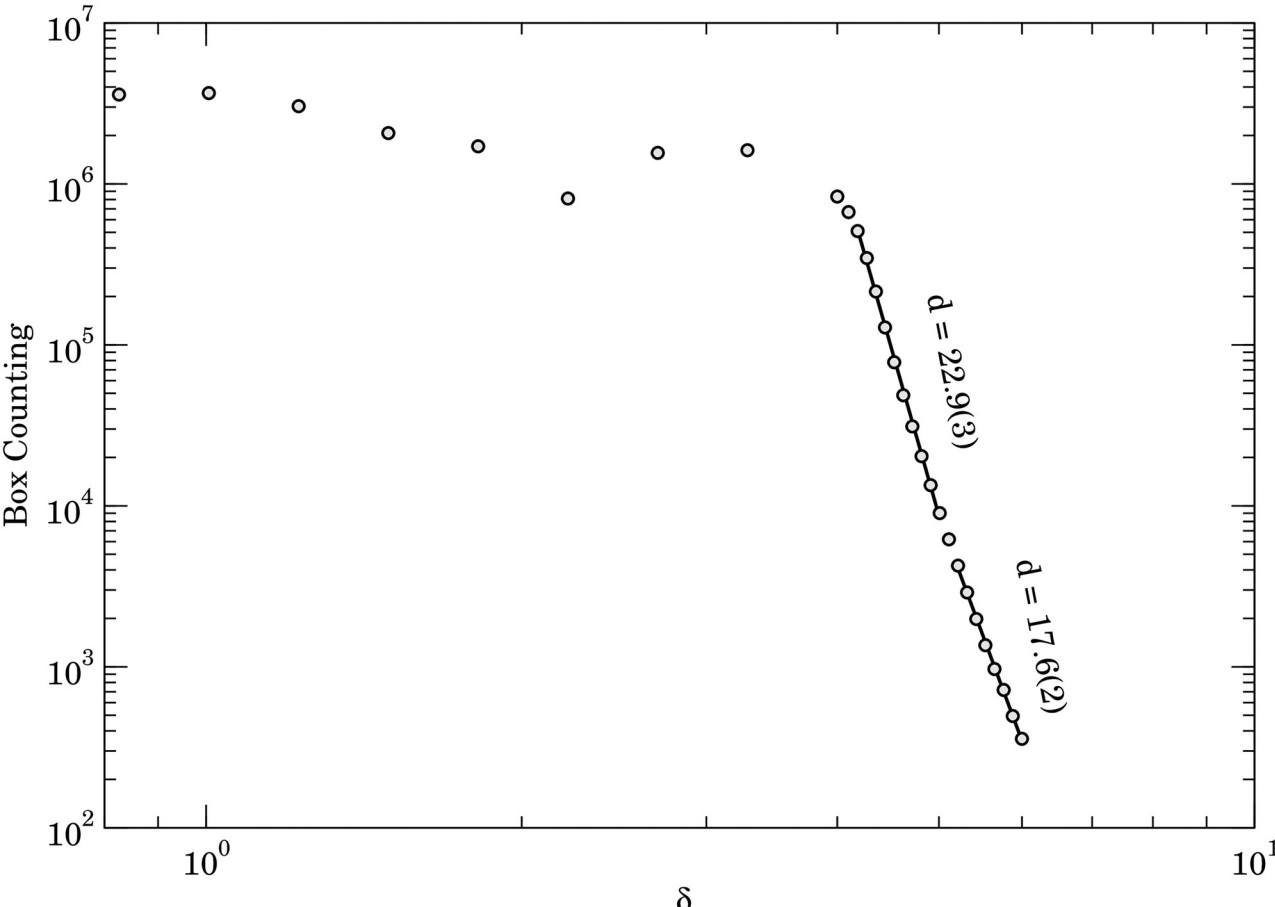

**Fig 6. Calculation of the fractal-dimension by the box-counting algorithm.**

other words, make a zoom in or out on it. So we approximate the structure mass for each scale $\delta$. Moreover, as discussed in the section on Fractals the relation between the mass and the scale is a power law whose exponent is the fractal dimension. Therefore, the fractal dimension can be calculated through a power-law regression of the number of boxes *versus* $\delta$ chart (see Fig 6).

So, we implemented the box-counting algorithm for a multi-dimensional space using parallel computing due to the high computational time demanded because of the space dimensionality and the number of points involved in the analysis.

## T-SNE dimension reduction

As the space used to represent the words in this study is 100-dimensional, it precludes the direct visualization of the data. So, we have used the t-distributed Stochastic Neighbor Embedding (t-SNE) algorithm to reduce the dimensionality of the space [44]. The advantage of this algorithm over others of this kind, such as PCA, is that it makes a nonlinear dimensionality reduction. So, it allows us to separate data in a way that we cannot do by using a straight line or a plane depending on the dimension of the reduced space. To describe more clearly how the t-SNE works, let us divide it into three steps as follows:

- The first step is calculating the Euclidean distances among all pairs of points in the dataset, considering all possible combinations. From these distances, the conditional probabilities

$p_{j|i}$ that $x_i$ will have $x_j$ as its neighbor is calculated as a Gaussian-shaped function of the distance $|x_i - x_j|$. Then, the t-SNE converts the conditional probability into the joint probability $p_{ij}$ of $i$ and $j$ being neighbours.

- Then, the t-SNE creates a point distribution in a low-dimension space (two or three) containing the same number of points of the original distribution. It may sound odd that this distribution is initially unrelated to the original dataset. However, the algorithm will use it to mimic the original distribution in the next step. This "artificial" data uses a t-student distribution instead of a Gaussian one.

- Finally, the Kullback-Leiber divergence [46], which measures how different two distributions are, is used to make the joint probability distribution of the data created in the second step as similar as possible to the one from the original dataset. At this moment, the application of artificial intelligence appears in the algorithm by using the gradient descent method with the cost function as the Kullback-Leiber divergence to minimize the difference between the joint probability distribution from the high-dimensional space and the one from the low-dimensional space

## Results

Once we created the vector representation of the word for the languages presented in Table 1, we applied the box-counting method to calculate the fractal dimension through the power-law regression of the relation between the number of boxes and the scale length $\delta$. Fig 6 shows this relation obtained for the English language. For small scales, whose length is considerably shorter than the typical distance among the points in the distribution, the y-axis value corresponds approximately to the number of types; therefore, it shows just minor changes in this range. For $\delta$ near two, the scale becomes similar to the typical distance in the points then a clearer relation appears for longer scales. In this longer-scale range, we identify two different power-law regimes: one with exponent 22.9±0.3 and another with exponent 17.6±0.2. So, we can conclude that the structure formed by English in the space created by the word2vec is multi-fractal with the two dimensions shown in Fig 6.

We also applied the box counting to the other languages concluding that all of them are related to multi-fractals with two different dimensions as shown in the values in Table 2 shows.

Subsequently, we calculated the type/token ratio (TTR). TTR is the ratio obtained by dividing the types (the total number of different words) occurring in a text or utterance by its tokens (the total number of words). A high TTR indicates a high degree of lexical variation while a low TTR indicates the opposite. This ratio is a relatively traditional and relevant measure in linguistics, and it expresses how often the same word is used in the language. So, a high ratio implies rich lexical density and variation in the sample and low ratios indicate low lexical variability. Then, we created a space formed by this ratio *versus* the longer-scale fractal dimension and spread the languages on it. Fig 7 shows the results where each circle represents a language. Our choice for analyzing the longer-scale fractal dimension due to longer scales in the word2-vec representation is based on broader structures in the language because it captures relations among words with more different uses in the language, since, in the space created by the word2vec, words that appear nearer have a more similar use in the sentences. But we also analyze the shorter-scale fractal dimension further.

It is significant that the fractal dimension is the exponent of the relation between the box number filled by the word vector representation and its size side. So, it is not an absolute measure that depends on the corpus size but a relative measure of how the number of filled

**Table 2. Two fractal-dimension obtained for each language.**

| Language | Shorter-scale dimension | Longer-scale dimension |
|---|---|---|
| English | 22.9 | 17.6 |
| Portuguese | 26.4 | 16.6 |
| German | 29.7 | 18.5 |
| Spanish | 22.9 | 16.4 |
| Russian | 28.3 | 16.4 |
| French | 26.9 | 17.1 |
| Chinese | 26 | 14.2 |
| Japanese | 26.4 | 16.7 |
| Korean | 22.3 | 16.7 |
| Italian | 24.4 | 15.5 |
| Arabic | 22.9 | 13.4 |
| Hebrew | 22.8 | 19.5 |
| Basque | 19.5 | 16.8 |
| Dutch | 20.8 | 18.2 |
| Swedish | 26.3 | 17.9 |
| Finnish | 24.4 | 20 |
| Estonian | 21.3 | 17.1 |

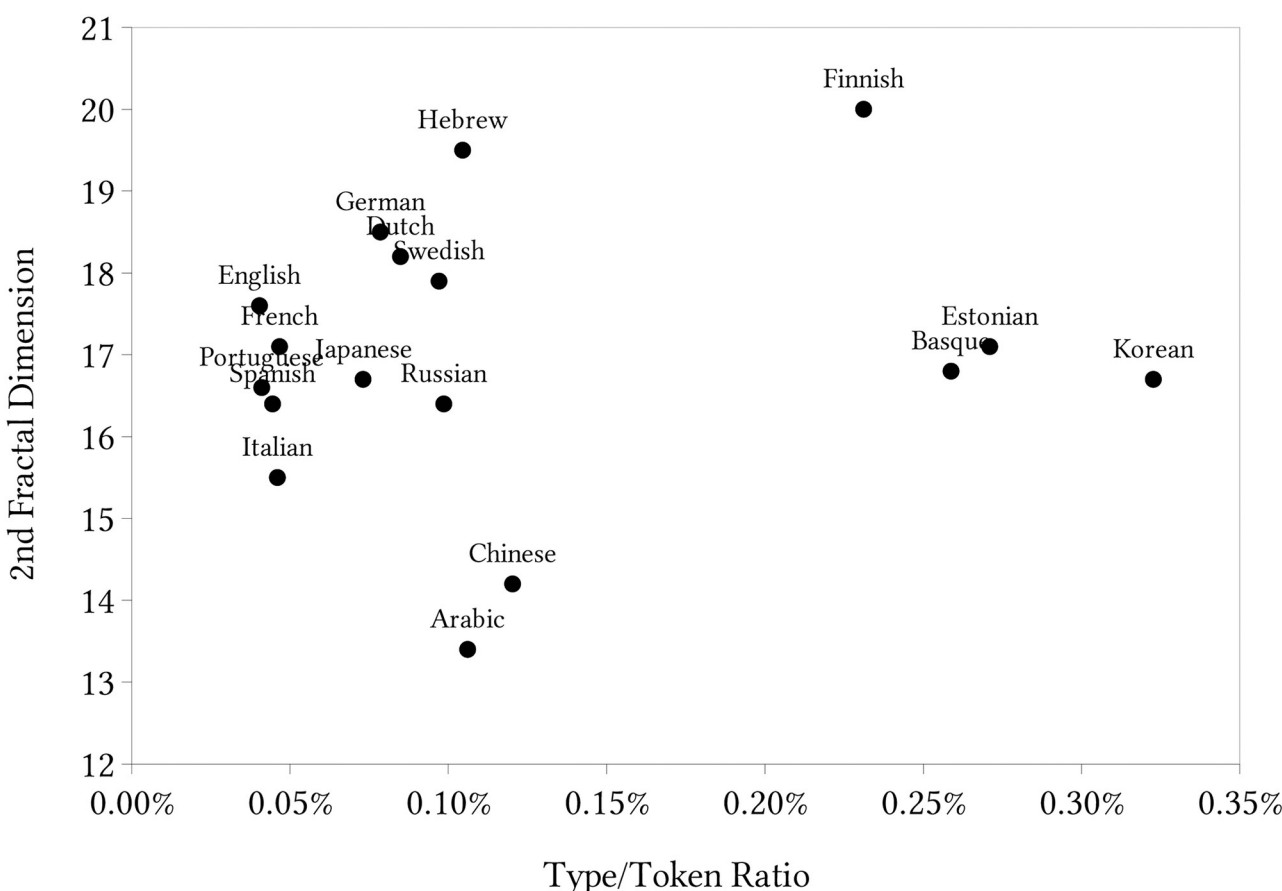

**Fig 7. Representation of the languages in a bi-dimensional space with the longer-scale fractal dimension.**

boxes changes with its size increase. As all our corpora are large enough (see Table 1), the fractal dimension converges to a value becoming independent of the corpus size. Therefore, the corpus-size difference among the languages does not influence the results shown in Table 2.

We can see some groups of languages that appear in proximity t each other, spontaneously forming clusters in the space. In the central-left part, we have French, Portuguese, Spanish and Italian all Romance languages in addition to English, showing the type-token ratio around 0.05% and the fractal dimension in the range from 15 to 18. It actually makes sense to see English, a Germanic language, clustered with Romance languages, given its history of contact with and borrowing extensively from both Latin and Norman French. The English vocabulary is estimated to have around 60% of its content based directly on borrowed Latin word forms or Latinate words which entered the language by way of Norman French. Slightly to the right and above the centre of the chart, we can see German, Dutch and Swedish with a lower repetition of words (type-token ratio from 0.07 to 0.10%) and higher fractal dimension (from 18 to 19). It is interesting that those three Germanic languages are pretty near in the tree by Levenshtein distance [46]. Below those languages, we have Russian, a Slavic Indo-European language and Japanese a language non-related to Indo-European and an apparent out of place language in the Indo-European clustering. Chinese and Arabic appear at the bottom with low token-type ratios and fractal dimensions. Here we face the only exception as to the association between the same historical root and a similar position in the chart: Hebrew and Arabic have rather similar token-type ratios but quite different longer-range fractal dimensions. However, considering the shorter-scale dimension, those languages became quite similar again. So, in this case, the same root left its fingerprint on the shorter-scale fractal dimension instead of the longer-scale one as the other languages. Finally, on the centre and right, we can see Estonian, Basque, and Korean quite near each other. As those languages do not share the same root, it would be relevant to study why their evolution makes them approximate in this space so that now they show a similar token-type ratio and longer-range dimension. On the other hand, Finnish, which is related to Estonian (Finno-Uralic languages), shares the same type/token range as Estonian but considerably higher longer-scale fractal dimension.

To deeply investigate the languages we cited above that were supposed to be close in the previous space because of the same root, but are not, we also create a space with the shorter-scaled fractal dimension and the TTR, that is shown in Fig 8. In this space, Hebrew and Arabic appear quite close to each other. Similar behavior occurs with Finish and Estonian, that share the same root, as they were apart in the longer-scale fractal dimension space but are nearer in this new space representation.

Therefore, for most of the languages we investigate, their natural evolution does not significantly alter their broader structures, indicating that they remain near their same-root languages in the longer-scale fractal dimension. The greater the fractal dimension, the greater the correlation between the elements that generated the fractal, in this case, the languages. As we are, at this moment, analysing the fractal dimension on longer scales, its value is related to the correlation between words or structures that are farther apart in the space generated by word2vec. In turn, due to the way word2vec generates the vector representation, closer points in this space have similar meanings or uses in the language. Therefore, what the fractal dimension at longer scales tells us is that there is a correlation between words or structures further apart in the word2vec representation that are associated with words or structures with more different meanings or uses in the language. However, for some languages, their natural evolution altered their broader structures but kept the structures related to words with more similar use (shorter distances in the word2vec representation) steadier. So, those languages moved apart in the longer-scale fractal dimension but remained close in the shorter-scale dimension.

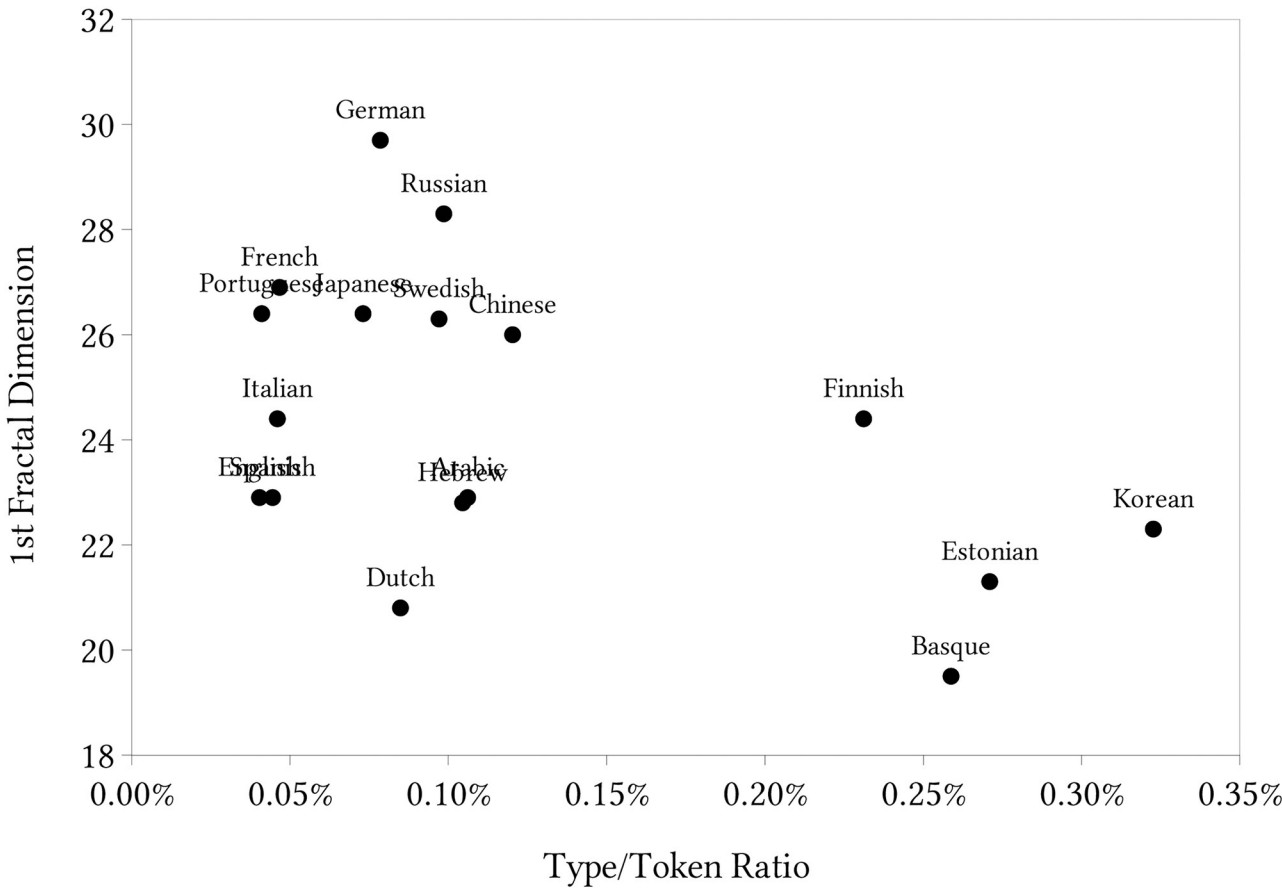

**Fig 8. Representation of the languages in a bi-dimensional space with the shorter-scale fractal dimension.**

As word2vec creates a high-dimension space, we cannot visualize the point distribution. So, by using the t-SNE (see section T-SNE dimension reduction) we decreased the space dimension from 100 to 2. Furthermore, we applied the Hoshen-Kopelman [47] algorithm to identify the cluster formed in this low-dimension space. Using the distance threshold of 0, 11 on the algorithm, we applied it to all languages of Table 2, but here we will focus on the illustrative cases of Russian and Hebrew. Fig 9 shows the Top-50 most prominent clusters identified for Russian and Fig 10 for Hebrew, where we highlighted the groups with different colours. The contrastive behaviour is pretty visual and clear: Russian has more homogeneous groups with the lower difference between the biggest and shortest ones, while Hebrew has quite heterogeneous groups with a considerable difference between the biggest and shortest ones.

From the clusters obtained above and in order to quantify the different behaviour that is now just visual, we create a chart setting the cluster position in the top-50 most prominent clusters on its horizontal axis and the size of the group on its vertical axis. Fig 11 shows this chart for Russian and Hebrew. Moreover, we also made a power-law regression obtaining the exponent −0.19 from Russian and −1.05 from Hebrew. As a result, the exponent in Hebrew that is more negative than the Russian one implies that Hebrew has more hierarchical groups of similar worlds.

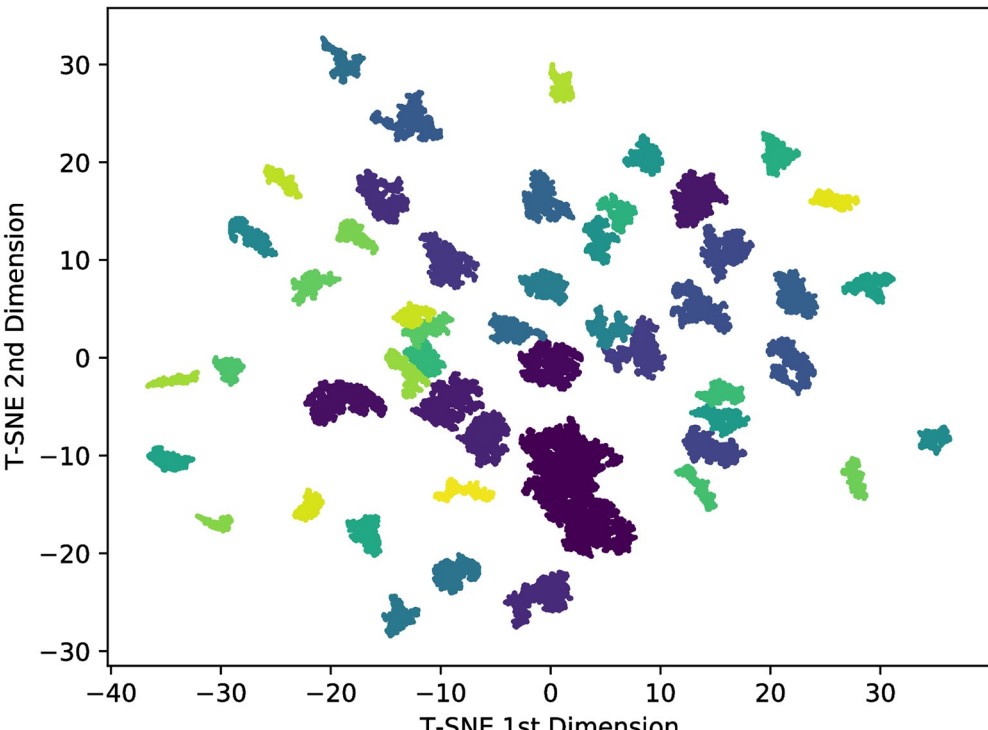

**Fig 9. Top-50 biggest clusters of Russian.**

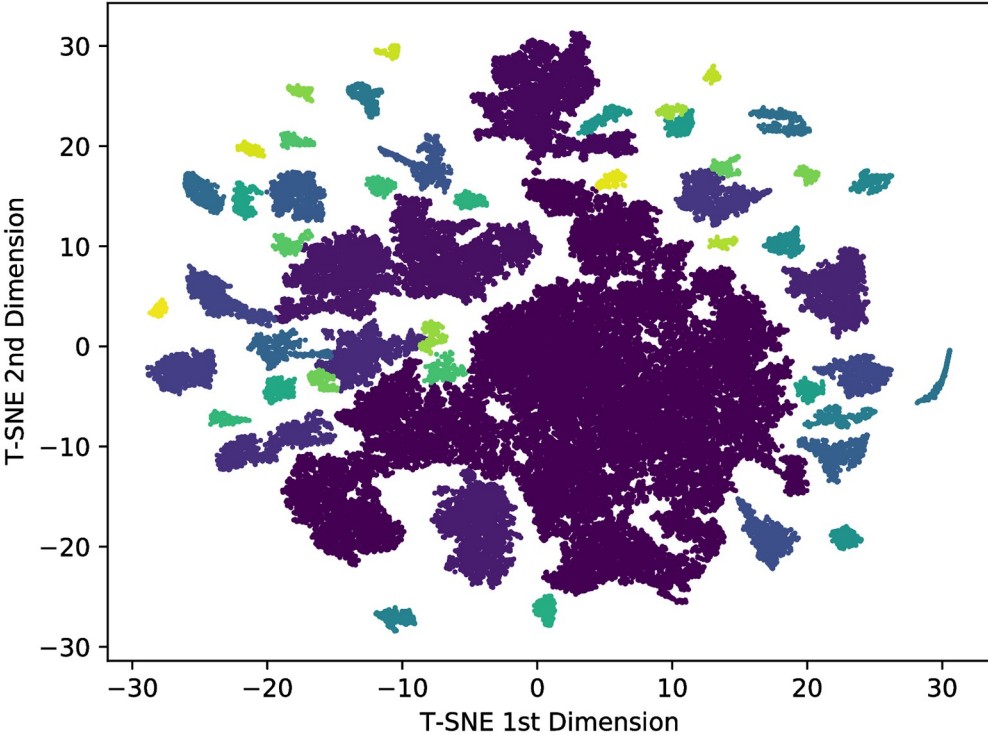

**Fig 10. Top-50 biggest clusters of Hebrew.**

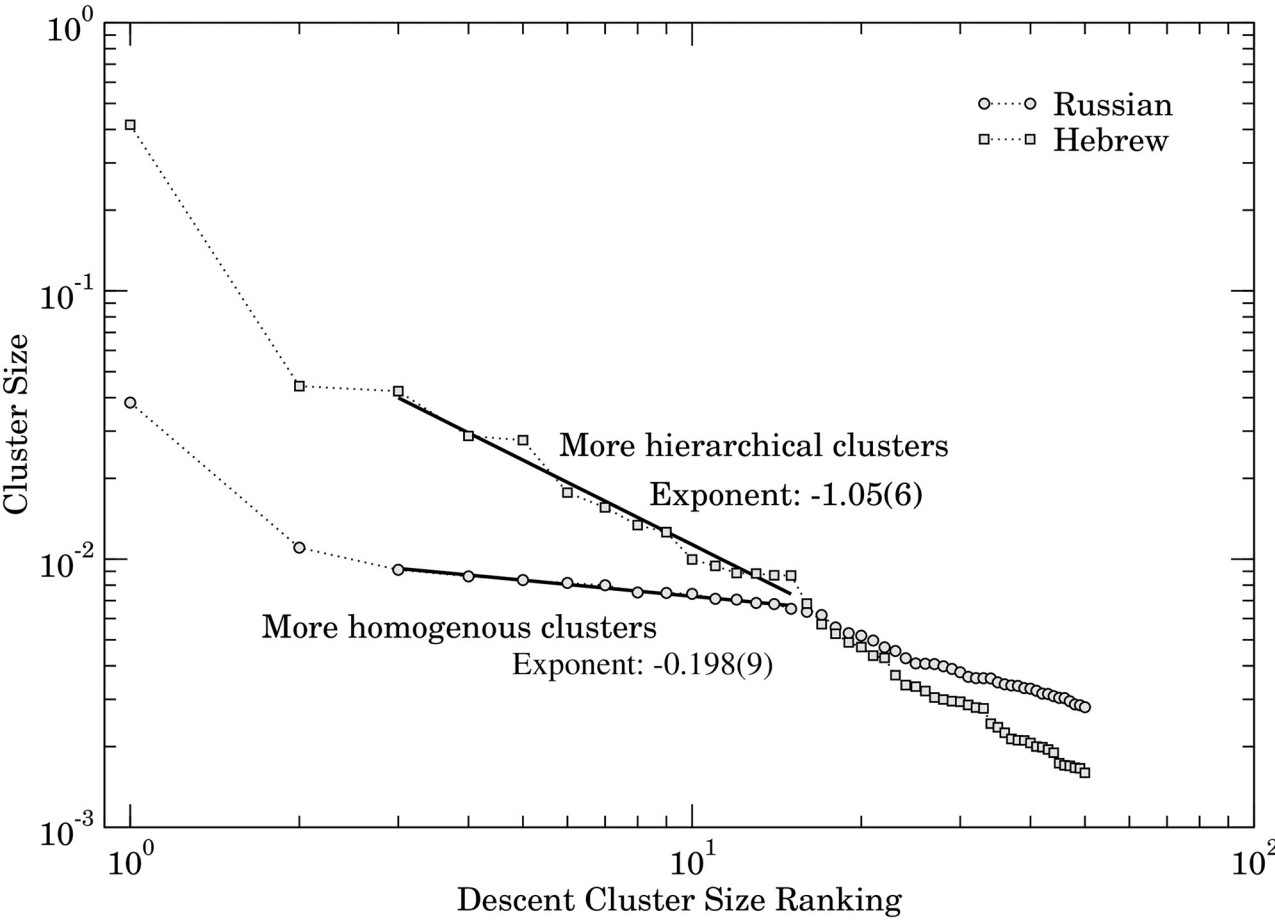

**Fig 11. Cluster size ranking.**

## Conclusion

We applied the word2vec algorithm to a representative corpus of several languages obtaining the vector representation of each language's words. This algorithm implements an embedding transformation to the corpus creating a vector representation that keeps the meaning relation among the words in this vector space.

Then, we calculated the shorter and longer-scale fractal dimensions of the languages in the vector space through the box-counting method. These fractal dimensions reveal how the points—representing words, in this case—fill the space. As the point positions in the word2vec space are related to their meaning and use in the language, the fractal dimension reflects the language structure with its semantic relations. Therefore, it allows a broader analysis of the language than staying restricted to the word formation [46].

When the longer-scaled fractal dimension and the Type-Token ratio are combined to create a bi-dimensional space that reflects the language structures, the languages create clusters according to the roots, with pretty few exceptions. The same root language appears grouped while languages from different roots stay apart.

As the languages evolve, an implication of the grouping above is that the part of language structure associated with the longer-scale fractal dimension is more stable or suffers similar alterations over time among the same root languages.

For the few exceptions mentioned above (Hebrew and Arabic), the longer-scale fractal dimension is not so similar. However, the shorter-scale one is, since those languages get near in the shorter-scale fractal dimension versus Type-Token ratio space.

So, for those specific languages, the part of language structure associated with the shorter-scale fractal dimension stays more stable or suffers similar alterations over time among the same root languages while the longer-scale fractal dimension changes.

Finally, we found that the linguistic interpretation of the fractal dimension is related to the homogeneity of the word group sizes formed in the vector representation of the word2vec. More heterogeneous cluster sizes are related to a higher fractal dimension, while clusters with more homogeneous cluster sizes are related to smaller fractal dimensions.

## Supporting information

**S1 File. Python code of the class Fractal that calculates the fractal dimension through the box counting.**
(PY)

**S2 File. Python code that implements the Hoshen Kopleman algorithm used to identify the clusters shown in Figs 9 and 10.**
(PY)

**S3 File. Python code to instantiate the class Fractal.**
(PY)

## Acknowledgments

The authors would like to thank the Editorial Coordinator of the review panel and the referees of this journal for their insightful criticisms, comments, and suggestions on an earlier version of this paper. This revised version owes a great deal to their input. Any remaining errors or shortcomings are solely the responsibility of the authors.

## Author Contributions

**Conceptualization:** Leonardo Costa Ribeiro, Américo Tristão Bernardes, Heliana Mello.

**Data curation:** Leonardo Costa Ribeiro, Américo Tristão Bernardes.

**Formal analysis:** Leonardo Costa Ribeiro, Américo Tristão Bernardes, Heliana Mello.

**Investigation:** Leonardo Costa Ribeiro, Américo Tristão Bernardes, Heliana Mello.

**Methodology:** Leonardo Costa Ribeiro, Américo Tristão Bernardes, Heliana Mello.

**Software:** Leonardo Costa Ribeiro.

**Supervision:** Américo Tristão Bernardes, Heliana Mello.

**Validation:** Leonardo Costa Ribeiro.

**Visualization:** Leonardo Costa Ribeiro.

**Writing – original draft:** Leonardo Costa Ribeiro, Américo Tristão Bernardes, Heliana Mello.

**Writing – review & editing:** Américo Tristão Bernardes, Heliana Mello.

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
