## [Decision Letter · Decision Letter 0]

14 Feb 2023

PONE-D-22-35269On the fractal patterns of language structuresPLOS ONE

Dear Dr. RIBEIRO,

Thank you for submitting your manuscript to PLOS ONE. After careful consideration, we feel that it has merit but does not fully meet PLOS ONE’s publication criteria as it currently stands. Therefore, we invite you to submit a revised version of the manuscript that addresses the points raised during the review process.

ACADEMIC EDITOR: The topic of the paper is very intriguing and makes a significant contribution to the study of languages. The authors should perhaps make more explicit than they already do how they used the CoNLL 2017 Corpora data (limited to selected languages) to respond to the first reviewer's comments. 

We look forward to receiving your revised manuscript.

Kind regards,

Ramona Bongelli, Ph.D.

Academic Editor

PLOS ONE

Journal Requirements:

5. Please ensure that you refer to Figures 2 and 3 in your text as, if accepted, production will need this reference to link the reader to the figure.

"This work was supported in part by the Brazilian agencies CNPq (307633/2019-5 and 312020/2021-0)."

Additional Editor Comments:

The topic of the paper is very intriguing and makes a significant contribution to the study of languages.

The authors should perhaps make more explicit than they already do how they used the CoNLL 2017 Corpora data (limited to selected languages) to respond to the first reviewer's comments.

Reviewers' comments:

Reviewer's Responses to Questions

**Comments to the Author**

1. Is the manuscript technically sound, and do the data support the conclusions?

Reviewer #1: No

Reviewer #2: Yes

2. Has the statistical analysis been performed appropriately and rigorously? 

Reviewer #1: I Don't Know

Reviewer #2: Yes

3. Have the authors made all data underlying the findings in their manuscript fully available?

Reviewer #1: No

Reviewer #2: Yes

4. Is the manuscript presented in an intelligible fashion and written in standard English?

Reviewer #1: No

Reviewer #2: Yes

5. Review Comments to the Author

Reviewer #1: I have to ask the authors to revise and resubmit their contribution. The reason is not that the article's findings are problematic (this is something I have not been able to assess yet) but that I cannot evaluate the manuscript at this stage, since the authors do NOT provide all the necessary data and code that one would need in order to replicate their results. Even if their study involves HPC computing, it is necessary that data AND code are shared already during the review stage with the reviewers and later -- in case of acceptance -- with the readers. The current MS claims to share data, but in the paper, I do not find a single link to their supplementary material, so I could not check the code for its suitability.

I will gladly provide a more thorough review of this study which may be potentially interesting, but only when data and code are present and allow me to inspect them thoroughly. This includes the code that was used to create the figures.

I recommend the authors to read the following two blog posts which will provide some guidance in how data and code can be shared when sending one's study to a scientific journal:

* https://calc.hypotheses.org/2782

* https://calc.hypotheses.org/2877

Reviewer #2: Things that need to be addressed as minor revisions:

Line #38: You mention phylogenetic methods, but I think a brief mention of what that means would be relevant here, particularly to help orient unfamiliar readers.

Line #66: I’d be interested in knowing what a ‘filament’ is.

Line #74: “discussed” should be “discuss”

Line #76: should it be: “the mathematical treatment of language”?

Line #104: Here you mention ‘long-range correlations’ for the first time, but you do not define what that means. This is important because you rely on this term (and on its opposite, short-range correlations) in the discussion of your findings.

Line #104: “can not be represented by a random walk” – I’m not sure what you mean here.

Lines #113-115: “The distribution of languages, their diversity and geographical coverage also present power-law behavior”

• I am not familiar with this. How is geographical coverage, for instance, governed by power laws?

Line #123: The Hurst exponent—you need to briefly mention what this is/does.

Lines #125-126: “This exponent is different from 0.5, which denotes that there are structures with long-range correlations.” Without further explanation of (a) what the exponent means, and (b) what long-range correlations are, this sentence does not make sense to the reader.

Line #203: “whose its elements” – remove ‘its’

Line #337: “We can see some groups of languages that spontaneously form in the space.” I’m not sure what you mean by ‘spontaneously form.’ Do you mean that have a tendency to cluster in Fig. 7?

Lines #370-372: “Therefore, for most of the languages we investigate, their natural evolution does not significantly alter their broader structures, indicating that they remain near their same-root languages in the longer-scale fractal dimension.” This is a very interesting claim, but I still wish that the concepts of shorter-scale and longer-scale fractal dimensions were more clearly discussed. That would help me better understand this idea.

Line #415: “For the few exceptions mentioned above…” Can you reiterate what those exceptions are here as a reminder?

Additional Comments

1. “Word embedding” is given as a keyword, but not discussed within the paper

2. Final paragraph of the introduction: It’s important that you start to discuss what you plan to do in this study, but some of this discussion is unclear. For example:

a. In lines #71 & 72, you write “we applied an algorithm developed in the last decade.” I immediately ask myself what the purpose/function of this algorithm is and where it came from.

b. Then you state that “we applied the tools of fractal geometry to observe similarities and differences among the many languages.” It’s not clear, however, how drawing on fractal geometry will allow you to observe these similarities and differences.

3. In the final paragraphs of your conclusion, you make some interesting claims related to the similarities in development of languages with similar/different roots. I’m happy to have learned this, but now I want to know why? What could be the motivator of this type of development? Is that discussed in the literature at all? What claims can you make?

6. PLOS authors have the option to publish the peer review history of their article (what does this mean?). If published, this will include your full peer review and any attached files.

Reviewer #1: **Yes: **Johann-Mattis List

Reviewer #2: No

---

## [Author Response · Author response to Decision Letter 0]

15 Apr 2023

Dear Ramona Bongelli,

Thank you for the opportunity to revise our manuscript and address the comments and concerns raised by the editor and the reviewers. We appreciate the time and effort that has been devoted to evaluating our work and providing us with constructive feedback.

In response to the comments raised by the editor and the reviewers, we have made significant revisions to our manuscript. We have carefully considered each of the issues raised, and we believe that our revisions have addressed all of the concerns that were raised. We have included additional data to support our findings, clarified our methods, and restructured the manuscript to improve its readability and coherence.

Regarding the concern raised by:

Journal Requirements:

Item 1.

Answer: We changed the symbol Yinyang for the Pilcrow. The references to figures were changed from Figure to Fig.

Item 2.

Answer: We submitted the codes with the reviewed manuscript. The files were named Fractal.py and Hoshen_Kopelman.py

Item 3.

Answer: We updated the information, and now the ‘Funding Information’ and ‘Financial Disclosure’ match.

Item 4.

Answer: The link to the data we used in the paper was added to the manuscript. Please see lines #247, 270 to 272 and references 40 and 42

Item 5.

Answer: Now, there the Fig 2 is refered in line 182 and Fig 3 in lines 218, 223.

Item 6.

Answer: We removed the funding information from Acknowledgments Section (line #457) and made the amended statement in the cover letter.

Item 7.

Answer: We checked the references and added some related to the data (see references 40 and 42). No articles we cited were retracted.

Reviewer #1

Answer: We explicitly added the link to the data on the manuscript. Please see lines 247, 270 to 272 and references 40 and 42. Moreover, we submitted jointly with the manuscript the code we used to make the analysis and the Figs (specifically 9 and 10). The files were named Fractal.py and Hoshen_Kopelman.py

Reviewer #2:

Item:

Line #38: You mention phylogenetic methods, but I think a brief mention of what that means would be relevant here, particularly to help orient unfamiliar readers.

Answer: We added an explanation regarding the phylogenetic methods. Please, see lines #41 to #53

Line #66: I’d be interested in knowing what a ‘filament’ is.

Answer: We changed the term filament by a clearer explanaition. See line #73

Line #74: “discussed” should be “discuss”

Answer: We replaced this sentece by a improved one to attempt to the "Additional Comments" #2.b. Please see lines #84 to 87.

Line #76: should it be: “the mathematical treatment of language”?

Answer: We corrected the sentence. Please see lines #87 and 88.

Line #104: Here you mention ‘long-range correlations’ for the first time, but you do not define what that means. This is important because you rely on this term (and on its opposite, short-range correlations) in the discussion of your findings.

Answer: We improved the sentence to clarify the meaning of ‘long-range correlations’. Please see lines #115 to 120.

Line #104: “can not be represented by a random walk” – I’m not sure what you mean here.

Answer: We improved the sentence to clarify the meaning of ‘random walk’. Please see lines #115 to 120.

Lines #113-115: “The distribution of languages, their diversity and geographical coverage also present power-law behavior”

• I am not familiar with this. How is geographical coverage, for instance, governed by power laws?

Answer: We improved the sentence to clarify it. Please see lines #128 to 130. But detailing what the papers we cited did, they showed that the number of different languages or diversity D in an area A increases with A as a power law D ~ A^z with z = 0.41±0.03.

Line #123: The Hurst exponent—you need to briefly mention what this is/does.

Answer: We added a brief explanation of the Hurst exponent. Please see lines #138 to 145.

Lines #125-126: “This exponent is different from 0.5, which denotes that there are structures with long-range correlations.” Without further explanation of (a) what the exponent means, and (b) what long-range correlations are, this sentence does not make sense to the reader.

Answer: We clarified the interpretation of Hust's exponent values (lines #138 to 145) and also an explanation of the long-range correlations (lines #115 to 120).

Line #203: “whose its elements” – remove ‘its’

Answer: We removed it. Please see line #225

Line #337: “We can see some groups of languages that spontaneously form in the space.” I’m not sure what you mean by ‘spontaneously form.’ Do you mean that have a tendency to cluster in Fig. 7?

Answer: Yes, we would like to refer to the clusters in Fig. 7. We improved the sentence to make it clearer. Please see lines #359 and 360.

Lines #370-372: “Therefore, for most of the languages we investigate, their natural evolution does not significantly alter their broader structures, indicating that they remain near their same-root languages in the longer-scale fractal dimension.” This is a very interesting claim, but I still wish that the concepts of shorter-scale and longer-scale fractal dimensions were more clearly discussed. That would help me better understand this idea.

Answer: The higher the fractal dimension, the higher the correlation between the elements that generated the fractal, in this case, the languages. As we are, at this moment, analyzing the fractal dimension on longer scales, its value is related to the correlation between words or structures that are farther apart in the space generated by word2vec. In turn, due to how word2vec generates the vector representation, closer points in this space have similar meanings or uses in the language. Therefore, what the fractal dimension at longer scales tells us is that there is a correlation between words or structures further apart in the word2vec representation that is associated with words or structures with more different meanings or uses in the language. We added this answer to the manuscript to clarify our claim. Please see lines #394 to #403.

Line #415: “For the few exceptions mentioned above…” Can you reiterate what those exceptions are here as a reminder?

Answer: We added the exceptions to this paragraph as a reminder. Please see line #446

Additional Comments

1. “Word embedding” is given as a keyword, but not discussed within the paper

Answer: We removed this keyword.

2. Final paragraph of the introduction: It’s important that you start to discuss what you plan to do in this study, but some of this discussion is unclear. For example:

a. In lines #71 & 72, you write “we applied an algorithm developed in the last decade.” I immediately ask myself what the purpose/function of this algorithm is and where it came from.

Answer: We used the word2vec because the vector representation it creates for the words brings their meaning. So, a general analysis for this space will consider a broader linguistic aspect of the language and not only how the words are formed as a sequence of letters as in previous works in the literature. We added this explanation to the manuscript. Please see lines #79 to 83.

b. Then you state that “we applied the tools of fractal geometry to observe similarities and differences among the many languages.” It’s not clear, however, how drawing on fractal geometry will allow you to observe these similarities and differences.

Answer: It is done by identifying the languages that appear in the same cluster (similar languages) or not (not so similar languages) in the space we create from the fractal dimension of the languages and their type$/$token ratio. We added this explanation to the manuscript. Please see lines #84 to 86.

3. In the final paragraphs of your conclusion, you make some interesting claims related to the similarities in development of languages with similar/different roots. I’m happy to have learned this, but now I want to know why? What could be the motivator of this type of development? Is that discussed in the literature at all? What claims can you make?

Answer: We are focused on it now to deeply understand the linguist reasons for the fractal dimension evolving over time. Moreover, to make an analysis based on data, we are analyzing the fractal dimension of the same language but getting corpora from different periods. Nevertheless, we will go deeper into it in further research.

We would like to thank the editor and the reviewers for their thoughtful comments and suggestions, which have helped us to improve our manuscript. We appreciate the opportunity to revise our work and believe that our revised manuscript is now ready for publication in your esteemed journal.

Sincerely,

Leonardo Costa Ribeiro

---

## [Editor Report · Decision Letter 1]

27 Apr 2023

On the fractal patterns of language structures

PONE-D-22-35269R1

Dear Dr. RIBEIRO,

We’re pleased to inform you that your manuscript has been judged scientifically suitable for publication and will be formally accepted for publication once it meets all outstanding technical requirements.

Kind regards,

Ramona Bongelli, Ph.D.

Academic Editor

PLOS ONE

---

## [Editor Report · Acceptance letter]

10 May 2023

PONE-D-22-35269R1 

On the fractal patterns of language structures 

Dear Dr. RIBEIRO:

I'm pleased to inform you that your manuscript has been deemed suitable for publication in PLOS ONE. Congratulations! Your manuscript is now with our production department. 

Kind regards, 

on behalf of

Professor Ramona Bongelli 

Academic Editor

PLOS ONE